# VEXAS Syndrome: Genetics, Gender Differences, Clinical Insights, Diagnostic Pitfalls, and Emerging Therapies

**DOI:** 10.3390/ijms26167931

**Published:** 2025-08-17

**Authors:** Salvatore Corrao, Marta Moschetti, Salvatore Scibetta, Luigi Calvo, Annarita Giardina, Ignazio Cangemi, Carmela Zizzo, Paolo Colomba, Giovanni Duro

**Affiliations:** 1Department of Health Promotion Sciences, Maternal and Infant Care, Internal Medicine and Medical Specialties (PROMISE), University of Palermo, 90133 Palermo, Italy; salvatore.corrao@unipa.it; 2Department of Internal Medicine, National Relevance and High Specialization Hospital Trust, ARNAS Civico, Di Cristina, Benfratelli, 90127 Palermo, Italy; salvatore.scibetta@arnascivico.it (S.S.); luigi.calvo@arnascivico.it (L.C.); annarita.giardina@arnascivico.it (A.G.); ignazio.cangemi@arnascivico.it (I.C.); 3Institute for Biomedical Research and Innovation (IRIB), National Research Council (CNR), 90146 Palermo, Italy; carmela.zizzo@irib.cnr.it (C.Z.); paolo.colomba@irib.cnr.it (P.C.); giovanni.duro@irib.cnr.it (G.D.)

**Keywords:** VEXAS syndrome, X-linked pathology, hematological and inflammatory manifestations, clinical features, lyonization affects, therapeutic approaches

## Abstract

VEXAS syndrome (Vacuoles, E1-enzyme, X-linked, Autoinflammation, and Somatic) is a recently identified late-onset autoinflammatory disorder characterized by a unique interplay between hematological and inflammatory manifestations. It results from somatic mutations in the *UBA1* gene, located on the short arm of the X chromosome. Initially, females were considered mere carriers, with the syndrome primarily affecting males over 50. However, recent evidence indicates that heterozygous females can exhibit symptoms as severe as those seen in hemizygous males. The disease manifests as systemic inflammation, macrocytic anemia, thrombocytopenia, chondritis, neutrophilic dermatoses, and steroid-dependent inflammatory symptoms. Due to its overlap with autoimmune and hematologic disorders such as relapsing polychondritis, Still’s disease, and myelodysplastic syndromes, misdiagnosis is common. At the molecular level, VEXAS syndrome is driven by impaired ubiquitination pathways, resulting in dysregulated immune responses and clonal hematopoiesis. A key diagnostic marker is the presence of cytoplasmic vacuoles in myeloid and erythroid precursors, though definitive diagnosis requires genetic testing for *UBA1* mutations. Traditional immunosuppressants and TNF inhibitors are generally ineffective, while JAK inhibitors and IL-6 blockade provide partial symptom control. Azacitidine and decitabine have shown promise in reducing disease burden, but hematopoietic stem cell transplantation (HSCT) remains the only curative treatment, albeit with significant risks. This review provides a comprehensive analysis of VEXAS syndrome, examining its clinical features, differential diagnoses, diagnostic challenges, and treatment approaches, including both pharmacological and non-pharmacological strategies. By enhancing clinical awareness and optimizing therapeutic interventions, this article aims to bridge emerging genetic insights with practical patient management, ultimately improving outcomes for those affected by this complex and often life-threatening disease.

## 1. Introduction

VEXAS syndrome, an acronym for Vacuoles, E1-enzyme, X-linked, Autoinflammatory, and Somatic, represents a groundbreaking discovery in adult-onset inflammatory diseases. First delineated in 2020 by Beck et al. [1], this syndrome is caused by somatic mutations in the *UBA1* gene, which encodes the ubiquitin-like modifier activating enzyme 1, a key regulator of protein ubiquitination processes. These mutations result in a complex clinical phenotype marked by severe systemic inflammation, hematologic abnormalities, and a profound impact on patients’ quality of life. In the past, VEXAS predominantly affects males over the age of 50 and frequently evades early recognition due to its overlapping features with other rheumatologic and hematologic conditions, leading to diagnostic delays and suboptimal management. Actually, this is not the case because we know that women can present the same clinical manifestations as men.

The clinical spectrum of VEXAS syndrome is remarkably heterogeneous, encompassing recurrent fevers, chondritis, neutrophilic dermatoses, and hematologic manifestations such as macrocytic anemia and thrombocytopenia [2]. A distinctive hallmark of the syndrome is the presence of cytoplasmic vacuoles in myeloid and erythroid precursors within the bone marrow, which serves as a critical diagnostic feature [3]. Due to its diverse clinical presentations, VEXAS syndrome is frequently mistaken for conditions such as adult Still’s disease, relapsing polychondritis, Sweet’s syndrome, and myelodysplastic syndromes, among other complex disorders. As a result, patients often undergo unnecessary and ineffective treatments before the true underlying genetic cause is correctly identified.

At the molecular level, the pathophysiology of VEXAS syndrome is intricately associated with the dysregulated function of the *UBA1* enzyme, leading to impaired protein homeostasis and abnormal activation of inflammatory pathways (Figure 1).

Recent research has clarified that monocytes from VEXAS patients show signs of functional exhaustion and altered chemokine receptor expression, which contribute to the ongoing inflammatory burden. Furthermore, inflammatory cytokine profiles in VEXAS demonstrate elevated levels of IL-6, IFN-γ, and TNF-α, linking it to other severe autoinflammatory syndromes [4]. These molecular findings enhance our understanding of disease mechanisms and identify potential therapeutic targets. Despite advancements in our understanding of its molecular underpinnings, therapeutic options remain limited. Conventional immunosuppressive regimens often yield only partial responses, necessitating high doses of corticosteroids to control symptoms. However, emerging treatments have demonstrated promising results in alleviating the disease’s inflammatory and hematologic aspects. Moreover, allogeneic hematopoietic stem cell transplantation is emerging as a potential curative intervention for selected patients, though it comes with significant risks and challenges. The role of targeted therapies, such as JAK inhibitors and IL-6 blockade, is currently under investigation and may offer more sustainable management strategies in the future [5]. This review aims to provide clinicians with a comprehensive and practical analysis of VEXAS syndrome, improving real-world diagnostic and therapeutic decision-making given the issue’s complexity. By exploring its clinical characteristics, diagnostic approach, differential diagnoses, applied clinical reasoning, and a critical update on therapeutic strategies and patient management, we intend to equip healthcare professionals with the essential tools to recognize, diagnose, and effectively manage this intricate condition.

Being still relatively unknown, it is frequently underdiagnosed due to limited clinical awareness and the restricted availability of specific diagnostic tests. Based on the previously described clinical manifestations, it is crucial for clinicians to consider initiating an appropriate diagnostic workup promptly in order to avoid misdiagnosis. Increased awareness of the disease has led to the development of specific diagnostic protocols, including Sanger sequencing of the *UBA1* gene. Targeted sequencing of the *UBA1* gene has proven to be a highly sensitive and efficient diagnostic tool for identifying pathogenic variants responsible for the clinical phenotype associated with VEXAS syndrome. However, despite the progress made, the genetic basis of a significant proportion of clinically relevant variants remains to be elucidated. Furthermore, the issue of genotype–phenotype correlation remains open and requires further investigation in large patient cohorts.

The protocol involves collecting peripheral blood in EDTA-containing tubes, which can be sent to specialized laboratories for analysis. Alternatively, Dried Blood Spot (DBS) samples can be used—an innovative technique involving the collection of blood drops on filter paper, which are then dried at room temperature and stored in moisture-protected conditions. This method offers significant advantages, including greater sample stability over time and the potential to simplify transport and storage logistics, particularly in resource-limited settings. The ongoing advancement of sequencing technologies and the implementation of increasingly sensitive and standardized protocols are essential tools to improve diagnostic capabilities and reduce delays in the diagnosis of VEXAS syndrome. These developments will contribute to a better understanding of the disease and the early identification of patients, ultimately having a positive impact on clinical management (Figure 2).

By synthesizing the latest evidence, we strive to enhance diagnostic precision, boost clinical awareness, and support the development of targeted therapeutic approaches, ultimately optimizing patient care and improving outcomes in this challenging and often fatal disease.

## 2. Genetics, Methodology, and Variants

VEXAS syndrome is a disorder of hematopoietic stem cells caused by mutations in the *UBA1* gene, located on chromosome Xp11.23, which codes for an enzyme involved in protein degradation [6]. The hypothesis of phenotypic differences between the sexes in this disease remains a topic of debate. To date, no studies have systematically compared clinical features between men and women. The underrepresentation of women in clinical and genetic studies has contributed to an incomplete understanding of the clinical course in women, which is often more nuanced but nonetheless clinically relevant [7]. Originally described as exclusive to men due to its genetic origin linked to the X chromosome, this disease is now also recognized in women, albeit still in rare cases. The clinical case reported by Barba et al. (2021) clearly demonstrates that the disease can manifest in a female patient, also with X monosomy that is, the loss of the second X chromosome, which would normally protect against the full expression of the *UBA1* gene mutation [8]. In that patient, the disease presented with severe and therapy-resistant symptoms, such as persistent fever, arthritis, auricular chondritis, and hematologic abnormalities typical of VEXAS.

This individual observation is confirmed by the comparative study of Echerbault et al. (2024), which analyzed data from 224 patients reported in the literature, including nine women [7]. The authors found that clinical, biological, and genetic features did not differ between males and females and that all affected women had X monosomy (constitutional or acquired). This suggests that although rarer, VEXAS in women follows the same clinical course and should be suspected using the same diagnostic criteria applied for men.

Supporting this, the systematic review by Loeza-Uribe et al. (2024) emphasizes that the disease falls within the group of autoinflammatory hematologic disorders, predominantly affecting adult males but also emerging in females when favorable genetic conditions, including X monosomy, are present [9]. The literature emphasizes that women with Turner syndrome, due to constitutional X monosomy, may be predisposed to developing VEXAS syndrome if they acquire a somatic mutation in *UBA1*. This identifies them as a female subgroup at risk, in whom clinical suspicion should be high in the presence of systemic inflammatory symptoms, hematological abnormalities, and resistance to conventional treatments. Therefore, they recommend including patients with Turner syndrome in the diagnostic and screening pathways for VEXAS.

Furthermore, the authors propose clinical suspicion criteria based on refractory inflammatory presentations associated with macrocytic anemia, which should include women, especially those over 50 years old.

Stubbins et al., (2022) report a case involving a female patient with Turner syndrome and a confirmed diagnosis of VEXAS [10]. This case enriches the understanding of female presentations of the disease and highlights the effectiveness of azacitidine as a bridging therapy prior to stem cell transplantation [10].

The convergent literature to date demonstrates that VEXAS syndrome is not exclusive to males and that, although rarer in females, it can manifest with an identical clinical spectrum, provided chromosomal abnormalities allow for the expression of the *UBA1* gene mutation (Table 1). It is therefore essential that clinicians include women in diagnostic pathways, especially in the presence of unexplained systemic inflammatory manifestations and hematologic disorders.

However, the fact remains that the clinical phenotype of VEXAS syndrome is broad and complex. A more detailed analysis of the molecular mechanisms underlying the disease may offer new perspectives for the development of precision therapeutic strategies aimed at acting on fundamental pathological processes. For this reason, many studies focus on understanding how molecular mechanisms influence disease onset and progression. Kusne et al. show that mutated stem cells progressively tend to replace healthy ones—lacking the mutation—through a process known as clonal hematopoiesis [11]. This evolutionary process may form either clinically irrelevant synonymous variants that may regress or to pathogenic variants such as ubiquitin-like modifier activating enzyme 1 (*UBA1*), which encodes ubiquitin ligase E1 and which may have a significant clinical implication. However, the causes and specific features of the clonal hematopoiesis underlying VEXAS syndrome are not yet fully understood. Consequently, severe inflammatory conditions occur, which mainly manifest themselves in adulthood. The main clinical presentations may lead to clinical diagnoses such as relapsing polychondritis, polyarteritis nodosa, Sweet’s syndrome, and myelodysplastic syndrome. Diagnosis requires an evaluation of the bone marrow to identify cytoplasmic vacuoles in myeloid and erythroid precursors. Nevertheless, the clinical presentation can be polymorphic and non-specific, leading to misdiagnosis or significant delays in identifying the disease. For this reason, genetic confirmation of *UBA1* gene mutations is essential. A promising approach involves a comprehensive analysis of *UBA1* mutations through gene sequencing. Since these mutations are acquired in the course of life and only present in a subset of hematopoietic cells, genetic analysis requires a particularly highly sensitive and precise methodology. Due to the complexity of the syndrome, some work reports the use of next-generation sequencing (NGS) to rapidly examine large numbers of DNA sequences in panels targeting multiple genes associated with hematological diseases. An example is a work carried out by Gutierrez-Rodrigues, exploring concomitant mutations in genes implicated in myelodysplastic syndromes and clonal hematopoiesis of indeterminate potential (CHIP), such as TET2 and DNMT3A [12]. However, whole-exome analysis, or multigenic panels, is often challenging due to the large number of variants not yet annotated in genomic databases. Moreover, these approaches involve long turnaround times and require bioinformatic support for result processing and interpretation. Other works, on the other hand, focusing on the clinical aspects of the syndrome, favor direct sequencing of the *UBA1* gene by Sanger sequencing, which proves particularly effective in identifying somatic mutations and small deletions in the *UBA1* gene [13]. This method can detect mutations with a lower detection sensitivity of approximately 15–20% mutant allele frequency. These mutations are primarily detectable in hematopoietic stem and progenitor cells, resulting in a mosaic population of altered myeloid cells. Boerger et al. developed a Digital Droplet PCR (ddPCR) method to detect seven *UBA1* mutations using four reactions and achieved an analytical sensitivity of 0.5% [14]. Despite these technological advances, the full spectrum of *UBA1* mutations and their phenotypic consequences remain incompletely understood. Future research employing longitudinal genomic analyses and functional assays is needed to elucidate the specific impact of different variants on disease progression and therapeutic response. Interestingly, mosaicism plays a crucial role in disease expression, as different mutant allele levels correlate with disease severity. These findings suggest that VEXAS may not be a distinct clinical entity but rather a manifestation within a broader clonal hematopoiesis disorder characterized by systemic inflammation. Considering this, single-cell RNA sequencing (scRNA-seq) is required to study this syndrome or to shed light on alterations in inflammatory pathways associated with disease pathogenesis [4].

## 3. VEXAS Syndrome in Women: The Role of Mosaicism and Co-Mutations

Although VEXAS syndrome was initially described as an exclusively male disease, emerging evidence has demonstrated that women can also develop severe manifestations, albeit at a lower frequency. This discrepancy is attributed mainly to X-chromosome mosaicism, a fundamental biological process in which one of the two X chromosomes in female cells undergoes random inactivation during embryonic development [14,15]. However, this inactivation is not always equal, and skewed X-inactivation can lead to a predominance of cells expressing the mutated *UBA1* allele, potentially driving disease pathogenesis in some women [4].

Recent genomic studies suggest that age-related clonal hematopoiesis [ARCH] and additional somatic mutations in genes such as TET2, DNMT3A, and ASXL1 may amplify disease severity in women with VEXAS [16,17,18]. These co-mutations, frequently associated with myeloid neoplasms and inflammatory disorders, may promote the expansion of pathological hematopoietic clones, leading to systemic inflammation and hematologic dysfunction [19]. Notably, in female patients with VEXAS, the interplay between skewed X-inactivation and clonal hematopoiesis results in substantial clinical heterogeneity, with some women exhibiting only mild inflammatory symptoms while others develop severe cytopenia, steroid-refractory inflammation, and progressive organ involvement [20]. Clinically, VEXAS in women often lacks the classical hallmarks observed in males, making diagnosis particularly challenging. Standard inflammatory markers, such as CRP and ferritin, may be inconsistently elevated, and bone marrow vacuolization is not always as pronounced. Consequently, many female patients endure prolonged diagnostic delays, often misclassified under autoimmune or hematologic disorders, without recognition of the underlying genetic etiology [21]. Given this complexity, systematic genetic screening for *UBA1* mutations should be considered in women presenting with unexplained cytopenia and refractory inflammation, particularly in those with concurrent somatic mutations indicative of clonal hematopoiesis. These insights underscore the need for further research into sex-specific disease mechanisms in VEXAS and refined diagnostic algorithms that incorporate X-inactivation patterns, hematopoietic clonal expansion, and molecular profiling. Recognizing the unique interplay between mosaicism and genetic co-mutations in female patients will be pivotal in advancing personalized therapeutic approaches and ensuring earlier intervention for this often-overlooked subset of VEXAS syndrome (Figure 3).

## 4. Clinical Manifestations and Diagnostic Challenges

VEXAS syndrome presents with a broad range of clinical manifestations, often resembling other inflammatory, autoimmune, and hematologic disorders (Table 2). The most common symptoms include recurrent fever, severe fatigue, weight loss, and systemic inflammation, frequently accompanied by chondritis, vasculitis, and neutrophilic dermatoses [2]. Cutaneous involvement is typical and varies from livedo reticularis to leukocytoclastic vasculitis, further complicating the clinical picture [1]. The hematologic features of VEXAS include macrocytic anemia, thrombocytopenia, and, in some cases, leukopenia. Cytoplasmic vacuoles in myeloid and erythroid precursors within the bone marrow remain a distinctive histopathologic marker [22]. However, these abnormalities are not exclusive to VEXAS and may overlap with myelodysplastic syndromes (MDS) and other clonal hematopoiesis disorders, which can lead to frequent misdiagnoses and inappropriate treatments [23]. The significant phenotypic overlap between VEXAS and conditions such as relapsing polychondritis, Sweet’s syndrome, Still’s disease, and vasculitides presents a major diagnostic challenge [24]. Patients are frequently treated with broad-spectrum immunosuppressants, corticosteroids, or biologics that target inflammatory cytokines, yet these therapies provide limited or temporary benefits [25]. The delay in obtaining an accurate diagnosis can lead to prolonged morbidity and unnecessary exposure to ineffective treatments.

Recent evidence indicates that VEXAS should be considered for any patient over 50 who presents with refractory inflammation, cytopenias, and steroid dependence, especially when standard autoimmune panels yield inconclusive results [1]. Genetic testing for *UBA1* mutations through next-generation sequencing is now recognized as the gold standard for definitive diagnosis, enabling more targeted and personalized therapeutic strategies [17].

## 5. Clinical Reasoning and Diagnostic Pitfalls

The diagnosis of VEXAS syndrome presents substantial challenges due to its clinical overlap with autoimmune, hematologic, and autoinflammatory disorders, which significantly increases the likelihood of misdiagnosis and inappropriate treatments. As highlighted by Corrao and Argano [26], clinicians frequently rely on pattern recognition heuristics, which, while efficient, can lead to cognitive biases such as anchoring bias, premature closure, and representativeness heuristic, mainly when dealing with a newly described entity like VEXAS. A crucial diagnostic pitfall arises from the phenotypic mimicry of VEXAS with diseases such as relapsing polychondritis, Still’s disease, myelodysplastic syndromes (MDS), large-vessel vasculitis, and ANCA-associated vasculitis. Many affected individuals exhibit chronic systemic inflammation, cytopenia, and steroid dependence, leading to their misclassification within established autoimmune or hematologic categories. This misclassification can be reinforced by confirmation bias, where clinicians selectively interpret findings that support their initial hypothesis, disregarding discordant features that could suggest an alternative diagnosis [27].

To overcome these diagnostic errors, it is essential to adopt a structured clinical reasoning framework, incorporating the following:Bayesian updating, where diagnostic probabilities are continuously revised based on emerging evidence [19].Dual-process thinking, balancing intuitive [System 1] rapid recognition with analytical [System 2] reasoning, particularly in cases with steroid-refractory inflammation and macrocytic anemia of unclear etiology [20].Cognitive forcing strategies, actively questioning initial diagnoses and considering alternative explanations to mitigate diagnostic inertia [20].

Another critical issue is the underutilization of genetic testing in older adults. Many clinicians mistakenly assume that monogenic disorders primarily manifest in pediatric populations. This misconception can lead to delayed recognition of *UBA1* mutations, particularly in male patients over 50 with chronic inflammation and macrocytic anemia. A structured diagnostic checklist incorporating cytopenia, vacuolated myeloid precursors, and systemic inflammation should serve as a clinical trigger for early genetic sequencing to confirm the diagnosis and avoid unnecessary immunosuppressive therapies [1]. By enhancing awareness of cognitive biases, integrating evidence-based heuristics, and prioritizing early genetic testing, clinicians can improve the accuracy of VEXAS syndrome diagnosis, ensuring more effective and individualized patient management (Table 3).

## 6. Therapeutic Approaches in VEXAS Syndrome

The management of VEXAS syndrome poses a considerable challenge due to the complex interaction between clonal hematopoiesis and systemic inflammation. Considering its diverse clinical presentation, therapeutic strategies need to be customized for each patient’s profile. Treatment approaches can be divided into ineffective or contraindicated therapies, partially effective methods, and emerging curative strategies, which include both pharmacological and non-pharmacological interventions.

### 6.1. Ineffective or Contraindicated Treatments

Many patients with VEXAS are initially treated with conventional immunosuppressants, particularly disease-modifying antirheumatic drugs (DMARDs) such as methotrexate, azathioprine, and mycophenolate mofetil. However, these agents fail to provide sustained disease control and have not shown significant effectiveness in modifying disease progression. Similarly, biologic agents that target TNF-α, such as infliximab and adalimumab, have produced disappointing results and should be avoided as monotherapy. Additionally, prolonged glucocorticoid therapy, while initially effective in controlling inflammation, is linked to severe adverse effects, including osteoporosis, diabetes, and an increased risk of infections, and should be used cautiously [21,22,23,24,25,26,27,28].

### 6.2. Partially Effective Approaches

Recent clinical experience indicates that Janus kinase (JAK) inhibitors (e.g., ruxolitinib, tofacitinib) may offer partial symptomatic relief, especially in decreasing systemic inflammation and cytopenia. However, their effectiveness varies, with many patients experiencing only transient or incomplete responses [29]. IL-6 inhibitors, such astocilizumab and sarilumab, have been used off-label, demonstrating moderate success in reducing inflammatory markers but often failing to completely control disease manifestations. Likewise, interleukin-1 (IL-1) inhibitors like anakinra and canakinumab have shown limited effectiveness, implying that IL-1-driven inflammation is not the main contributor to VEXAS pathogenesis [23].

### 6.3. Emerging Curative and Targeted Therapies

Since VEXAS is fundamentally a clonal hematopoiesis disorder, therapies targeting hematopoietic progenitor cells have emerged as the most promising interventions. Hypomethylating agents, particularly azacitidine and decitabine, have shown efficacy in reducing the inflammatory burden and improving hematologic parameters, making them viable for patients with concomitant myelodysplastic features [30]. The only known curative therapy remains allogeneic HSCT, which can eradicate mutant hematopoietic clones and provide long-term remission. However, HSCT carries significant risks, including graft-versus-host disease (GVHD) and high treatment-related mortality, and is therefore recommended only for carefully selected patients with severe or refractory disease [23,24].

### 6.4. Non-Pharmacologic and Supportive Care Strategies

A multidisciplinary approach is crucial for managing VEXAS, necessitating collaboration among rheumatologists, hematologists, and immunologists. Supportive care measures include transfusion support for anemia, infection prophylaxis for immunosuppressed patients, and osteoporosis prevention for those on long-term corticosteroids. Moreover, emerging research is investigating the role of metabolic and oxidative stress pathways as potential therapeutic targets, though these remain in experimental stages [1]. In conclusion, VEXAS syndrome presents a complex therapeutic dilemma. Conventional immunosuppressants are largely ineffective, and emerging targeted therapies offer varying degrees of success. While JAK inhibitors, IL-6 blockade, and hypomethylating agents provide partial benefit, allogeneic hematopoietic stem cell transplantation remains the only definitive cure. Future efforts should focus on precision medicine approaches, novel targeted therapies, and refining patient selection for HSCT, ultimately improving outcomes in this severe and often life-threatening disorder.

The treatment of VEXAS syndrome remains an unresolved clinical challenge, as no effective and universally recognized therapy currently exists to durably modify the course of the disease. While glucocorticoids represent the most effective initial strategy, their benefit is transient and often associated with chronic dependency and serious complications, whereas conventional and biologic immunosuppressive therapies yield inconsistent or disappointing results, offering only partial and frequently temporary improvements. Targeted therapies, such as JAK inhibitors or hypomethylating agents, although promising in selected subgroups, have not yet demonstrated definitive disease-modifying effects, nor are there established criteria to guide their use. Allogeneic hematopoietic stem cell transplantation, theoretically curative, is limited to a small number of eligible patients and carries significant risks. In this context, disease management remains essentially symptomatic and palliative, with high inter-patient variability. Nevertheless, the intense momentum of scientific research—which has already elucidated the genetic and molecular basis of the syndrome in just a few years—offers a realistic hope that more effective, personalized, and potentially curative treatments may emerge in the near future.

## 7. Conclusions

VEXAS syndrome, caused by somatic mutations in the *UBA1* gene, leads to severe inflammatory conditions that manifest in adulthood. Its genetic origin, linked to the X chromosome, highlights a significant clinical impact even in female patients, who remain underrepresented in research due to a frequently milder disease course influenced by lyonization. The primary manifestations include systemic inflammation and hematologic abnormalities. However, the clinical phenotype is broad and complex, often resulting in misdiagnosis, diagnostic delays, and initially inappropriate treatments. A deeper understanding of the molecular mechanisms underlying this syndrome could pave the way for precision therapeutic strategies that target the disease at its core. One promising avenue is the longitudinal study of *UBA1* mutations through genomic sequencing and functional assays, which will be essential for elucidating the impact of different variants on disease progression and treatment response. Another potential approach is to disrupt the inflammatory cascade associated with the disease by targeting the ubiquitination pathway. Additionally, restoring *UBA1* function could represent a significant therapeutic breakthrough in re-establishing cellular homeostasis. With these strategies, we aim to provide clinicians with concrete tools to enhance the recognition and management of VEXAS syndrome, ensuring timely diagnosis and personalized care. The role of mosaicism and lyonization in female patients further underscores the complexity of the disease and the necessity for refined diagnostic approaches to detect milder cases. Future studies should concentrate on unraveling the impact of mosaicism in women, seeking to identify subtle clinical manifestations and optimize patient outcomes. A deeper understanding of these mechanisms could ultimately elevate quality of life and improve prognosis for individuals affected by this complex and potentially life-threatening condition.

This remains a significant challenge owing to the involvement of multiple organ systems, the variability in clinical manifestations, the common co-occurrence with hematologic malignancies, the reliance on glucocorticoids, and the limited response to standard immunosuppressive treatments.

## Figures and Tables

**Figure 1 ijms-26-07931-f001:**
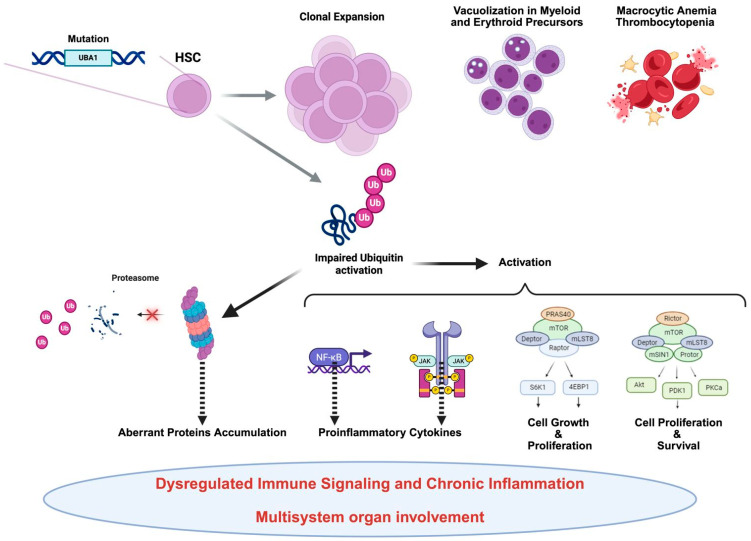
This figure illustrates how *UBA1* mutations disrupt immune homeostasis by altering ubiquitin activation in hematopoietic stem cells. This dysfunction leads to clonal expansion and the development of pathological features such as macrocytic anemia, cytoplasmic vacuoles, and systemic inflammation. At the molecular level, aberrant activation of key signaling pathways—namely NF-κB, JAK-STAT, and mTOR—drives chronic inflammation through excessive cytokine production, prolonged inflammatory signaling, and metabolic dysregulation. These processes collectively result in steroid-dependent inflammation, neutrophilic dermatoses, and multi-organ involvement, contributing to resistance to conventional immunosuppressive therapies. This visual representation provides a comprehensive overview of the pathophysiological mechanisms underlying VEXAS syndrome, offering insights into its impact on immune homeostasis and hematopoiesis.

**Figure 2 ijms-26-07931-f002:**
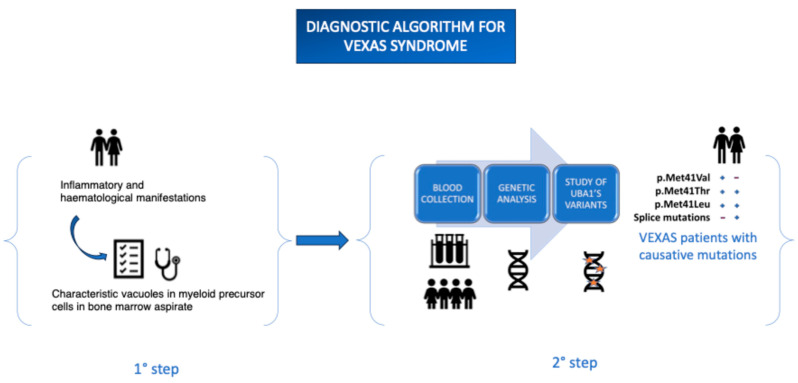
Proposed diagnostic workflow for VEXAS syndrome. The process is structured in two main phases. The first step involves clinical suspicion based on characteristic signs and symptoms (e.g., persistent fever, chondritis, cytopenia, macrocytic anemia, and characteristic vacuoles in myeloid precursor cells in bone marrow aspirate). The second step includes blood collection, genetic analysis, and the study of *UBA1* gene variants, which confirms the diagnosis. The “+” and “−” signs indicate the relative frequency of mutations (literature data) in male and female patients. This pathway helps guide the diagnostic evaluation, especially in adult patients with systemic inflammation and hematologic abnormalities.

**Figure 3 ijms-26-07931-f003:**
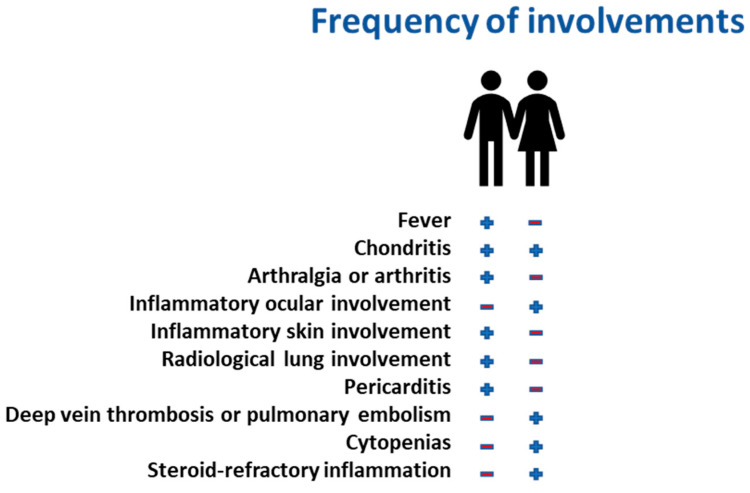
Comparing clinical features between males and females.

**Table 1 ijms-26-07931-t001:** An overview of published studies reporting VEXAS syndrome in female patients. The table summarizes the literature on VEXAS syndrome in women, detailing three key studies that document clinical cases and cohorts, highlight the role of X monosomy, and discuss diagnostic and screening implications specific to female patients.

Study	Focus	Findings in Women	Implications
Barba et al., 2021 [8]	Single clinical case of VEXAS in a woman with X monosomy	Severe symptoms in a woman with X monosomy; confirms clinical expression in females	VEXAS can clinically manifest in women under specific genetic conditions
Echerbault et al., 2024 [7]	Comparative study of 224 cases (215 men, 9 women)	No difference in clinical/biological features vs. men; all affected women had X monosomy	Diagnostic criteria should be the same for men and women
Loeza-Uribe et al., 2024 [9]	Systematic review on pathogenesis, diagnosis, and treatment	VEXAS can emerge in women with favorable genetics (e.g., X monosomy); proposes screening criteria	Women should be included in screening, especially if over 50 with macrocytic anemia

**Table 2 ijms-26-07931-t002:** Spectrum of clinical manifestations and their frequency across various categories.

Category	Symptom/Sign	Frequency
Systemic	Recurrent fever	Very common
Systemic	Severe fatigue	Very common
Systemic	Weight loss	Common
Hematologic	Macrocytic anemia	Very common
Hematologic	Thrombocytopenia	Common
Hematologic	Leukopenia	Less common
Hematologic	Neutropenia	Less common
Hematologic	Clonal haematopoiesis	Common
Inflammatory	Steroid-dependent inflammation	Very common
Inflammatory	Elevated inflammatory markers [CRP, ESR]	Very common
Inflammatory	Elevated ferritin levels	Common
Inflammatory	Cytokine dysregulation [IL-6, TNF-α, IFN-γ]	Very common
Inflammatory	Monocyte and neutrophil dysfunction	Less common
Cutaneous	Neutrophilic dermatoses [Sweet’s syndrome, leukocytoclastic vasculitis]	Common
Cutaneous	Lived oreticularis	Common
Cutaneous	Recurrent skin ulcers	Less common
Cutaneous	Purpura	Less common
Pulmonary	Pulmonary inflammation [interstitial lung disease]	Less common
Pulmonary	Pleural effusion	Less common
Pulmonary	Alveolar hemorrhage	Rare
Rheumatologic	Relapsing polychondritis [auricular, nasal, tracheal involvement]	Common
Rheumatologic	Joint pain [arthritis, arthralgia]	Common
Rheumatologic	Synovitis	Less common
Cardiovascular	Aortitis	Less common
Cardiovascular	Vasculitis [large and medium vessels]	Less common
Gastrointestinal	Gastrointestinal inflammation [colitis, enteritis]	Less common
Gastrointestinal	Hepatosplenomegaly	Less common
Neurologic	Peripheral neuropathy	Rare
Neurologic	Cognitive dysfunction	Rare
Diagnostic hallmark	Bone marrow vacuolization in myeloid and erythroid precursors	Diagnostic hallmark
Diagnostic hallmark	Somatic mutation in *UBA1* gene	Diagnostic hallmark

**Table 3 ijms-26-07931-t003:** The main clinical red flags associated with VEXAS syndrome. The table summarizes key demographic, clinical, hematologic, and radiologic features that should raise suspicion for VEXAS syndrome, especially in adult patients presenting with systemic inflammation, cytopenia, and treatment-resistant symptoms. Data are based on published cohort studies and expert reviews [9,20].

Category	Manifestations/Red Flag	Frequency/Notes
Demographic profile	Age > 50 years	Higher prevalence in men; rare female cases associated with monosomy X
Systemic symptoms	Recurrent fever, fatigue, night sweats, weight loss	Very common (>65%)
Hematologic abnormalities	Macrocytic anemia, thrombocytopenia, neutropenia, bone marrow vacuolization	Nearly universal (96–100%); recurrent cytopenias and macrocytosis
Cutaneous involvement	Neutrophilic dermatosis (Sweet-like), leukocytoclastic vasculitis, erythematous plaques, periorbital edema	Up to 100%; skin biopsies often show *UBA1*-mutated clonal infiltrates
Ocular involvement	Orbital edema, scleritis, uveitis, conjunctivitis	Sometimes misdiagnosed as periorbital cellulitis
Thrombotic events	Venous thromboses (DVT/PE), rare arterial events	~35–56%; often unprovoked and in presence of elevated FVIII or antiphospholipid antibodies
Estimated prevalence	1/4269 men > 50 years; 1/13,591 in the general population	Higher than some systemic vasculitides; similar to myelodysplastic syndromes

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
