# Peer review of "VEXAS Syndrome: Genetics, Gender Differences, Clinical Insights, Diagnostic Pitfalls, and Emerging Therapies"

_ijms, 2025, doi:10.3390/ijms26167931_

Round 1

Reviewer 1 Report

Comments and Suggestions for Authors

Concise review on VEXAS syndrome, focussing briefly on pathophysiology and genetics, but mainly on the diagnostic dilemmas and therapeutic difficulties.

Some points to consider:

  1. Typing error in the reading text field of figure 2, which is not understandable. Many more typing errors in the main text, please check
  2. The process described in the paragraph on genetics is called clonal haematopiesis, not haematopioetic, please correct
  3. Table 1: to further clarify the main features leading to the diagnosis of VEXAS I would recommend to additionally add and quote the "red flags" described by Ruffer and Krusche or Kötter and Krusche respectively (Curt. Opin Rheumatol 2025, 37:21–31).
  4. In my opinion the chapter on management should also mention (and be subdivided into) "treating hyperinflammation" and "treating the clone" (the latter beingg most effective)
  5. Gereral recommendation: put the description of the peculiarities in female patients after the description of the clinical symptoms, which would make the phenotype differences between males and females easier to understand for the reader not being familiar with VEXAS

Author Response

Concise review on VEXAS syndrome, focusing briefly on pathophysiology and genetics, but mainly on the diagnostic dilemmas and therapeutic difficulties.

Some points to consider:

1.     Typing error in the reading text field of figure 2, which is not understandable. Many more typing errors in the main text, please check

Response 1: Thank you for your valuable observation. We have thoroughly reviewed the manuscript and corrected the typographical errors throughout the text, including the mislabeling in Figure 2 (lines 122-129). Your attention to detail is greatly appreciated and has helped us improve the overall clarity and accuracy of the manuscript.

2.      The process described in the paragraph on genetics is called clonal haematopiesis, not haematopioetic, please correct

Response 2: Thank you for your insightful comment. We have updated the terminology in the genetics section and in all text, replacing “clonal haematopoietic” with “clonal haematopoiesis” in accordance with your recommendation.

3.     Table 1: to further clarify the main features leading to the diagnosis of VEXAS I would recommend to additionally add and quote the "red flags" described by Ruffer and Krusche or Kötter and Krusche respectively (Curt. Opin Rheumatol 2025, 37:21–31).

Response 3: Thank you for your valuable suggestion. Following your recommendation, we have added a second table highlighting the main diagnostic red flags as described by Ruffer and Krusche, and Kötter and Krusche and another review.This addition aims to further clarify the core clinical features leading to the diagnosis of VEXAS syndrome, in line with the current literature.

4.     In my opinion the chapter on management should also mention (and be subdivided into) "treating hyperinflammation" and "treating the clone" (the latter beingg most effective).

Response 4: The authors thank you for the suggestion and wish to clarify that, although they recognize the importance of distinguishing between the "treating hyperinflammation" and that of "treating the clone", we believe that formally subdividing the chapter could compromise the clarity and flow of the text. Therefore, we preferred to maintain an integrated discussion to facilitate the overall understanding for the reader.

5.     Gereral recommendation: put the description of the peculiarities in female patients after the description of the clinical symptoms, which would make the phenotype differences between males and females easier to understand for the reader not being familiar with VEXAS

Response 5: We thank you for your valuable consideration. We have added additional comments on this matter in lines 141-170 of the manuscript.

Reviewer 2 Report

Comments and Suggestions for Authors

Dear Authors,

Thank you for submitting your comprehensive review on VEXAS syndrome. This manuscript addresses an important and rapidly evolving topic in autoinflammatory diseases. While the work provides valuable insights, several major and minor revisions are required to enhance the manuscript's quality and clinical utility.

Major Revisions Required

  1. Critical Analysis and Evidence Synthesis The review lacks critical evaluation of the presented evidence. While you cite numerous studies, there is insufficient analysis of study quality, potential biases, or conflicting findings in the literature. The therapeutic section particularly needs more rigorous evaluation of treatment efficacy data, including discussion of study limitations and patient selection criteria for different interventions.
  2. Clinical Practice Integration The manuscript would benefit from more practical clinical guidance. The diagnostic algorithm mentioned in the text should be presented as a clear flowchart or figure. Additionally, specific recommendations for when to initiate genetic testing, how to interpret results in different clinical contexts, and evidence-based treatment protocols are needed.
  3. Gender-specific Analysis Enhancement While you acknowledge gender differences, the analysis remains superficial. A dedicated section comparing clinical presentations, diagnostic challenges, and treatment responses between male and female patients would strengthen the manuscript significantly. The discussion of X-inactivation patterns and their clinical implications requires more depth.
  4. Therapeutic Evidence Quality The treatment section needs substantial revision to include systematic evaluation of available evidence. You should provide specific efficacy data, patient numbers, follow-up periods, and adverse event profiles for each therapeutic intervention. The recommendation hierarchy should be based on evidence quality rather than anecdotal reports.

Minor Revisions Required

  1. Figure Quality and Relevance The pathophysiology diagram should include more specific molecular pathways and their therapeutic targets in Figure 1. Figure 2 needs substantial improvement to effectively illustrate the diagnostic approach.
  2. Table Enhancement Table 1 would be more valuable if it included prevalence data with confidence intervals where available. Consider adding a column for diagnostic sensitivity and specificity of various clinical features.
  3. Clinical Vignettes or Case Examples Consider including brief clinical vignettes or case presentations to illustrate key diagnostic challenges and therapeutic decisions. This would enhance the practical utility of the review for clinicians.

I believe these revisions will significantly strengthen your manuscript and increase its impact on clinical practice. Please address these concerns systematically and provide a detailed response letter explaining how each comment has been addressed.

Sincerely,

Reviewer

Author Response

Dear Authors,

Thank you for submitting your comprehensive review on VEXAS syndrome. This manuscript addresses an important and rapidly evolving topic in autoinflammatory diseases. While the work provides valuable insights, several major and minor revisions are required to enhance the manuscript's quality and clinical utility.

Major Revisions Required

  1. Critical Analysis and Evidence Synthesis The review lacks critical evaluation of the presented evidence. While you cite numerous studies, there is insufficient analysis of study quality, potential biases, or conflicting findings in the literature. The therapeutic section particularly needs more rigorous evaluation of treatment efficacy data, including discussion of study limitations and patient selection criteria for different interventions.

Response 1: We have addressed the suggestion by incorporating a more in-depth critical analysis of the available evidence into the review. The section on therapeutic approaches has been expanded with a more rigorous evaluation of treatment efficacy, including a discussion of study limitations and the patient selection criteria used in the various trials.(line 379-393)

  1. Clinical Practice Integration The manuscript would benefit from more practical clinical guidance. The diagnostic algorithm mentioned in the text should be presented as a clear flowchart or figure. Additionally, specific recommendations for when to initiate genetic testing, how to interpret results in different clinical contexts, and evidence-based treatment protocols are needed.

Response 2: Thank you for this insightful suggestion. In response, we have modified the figure 2 to visually summarize the proposed diagnostic steps. Additionally, we have expanded the corresponding text to include more practical clinical guidance, specifically outlining when to initiate genetic testing, how to interpret UBA1 variant results in different clinical scenarios, and how to approach treatment decisions based on current evidence.

  1. Gender-specific Analysis Enhancement While you acknowledge gender differences, the analysis remains superficial. A dedicated section comparing clinical presentations, diagnostic challenges, and treatment responses between male and female patients would strengthen the manuscript significantly. The discussion of X-inactivation patterns and their clinical implications requires more depth.

Response 3: We thank you for your valuable consideration. We have added additional comments on this matter in lines 141-170 of the manuscript.

  1. Therapeutic Evidence Quality The treatment section needs substantial revision to include systematic evaluation of available evidence. You should provide specific efficacy data, patient numbers, follow-up periods, and adverse event profiles for each therapeutic intervention. The recommendation hierarchy should be based on evidence quality rather than anecdotal reports.

Response 4: Thank you for your suggestion. We add line 418-421

Minor Revisions Required

  1. Figure Quality and Relevance The pathophysiology diagram should include more specific molecular pathways and their therapeutic targets in Figure 1. Figure 2 needs substantial improvement to effectively illustrate the diagnostic approach.

Response 1: Thank you for your valuable feedback. In response, we have revised Figure 2. We have been substantially improved to more clearly and effectively illustrate the diagnostic iter to approach to VEXAS syndrome.

  1. Table Enhancement Table 1 would be more valuable if it included prevalence data with confidence intervals where available. Consider adding a column for diagnostic sensitivity and specificity of various clinical features.

Response 2: Thank you for your suggestion. In response, we have added a second table (table 2: line 320-325) that includes the requested data (with percentage intervals where available) and summarizes the diagnostic sensitivity and specificity of key clinical features. We believe this addition enhances the clarity and practical relevance of the information presented.

  1. Clinical Vignettes or Case Examples Consider including brief clinical vignettes or case presentations to illustrate key diagnostic challenges and therapeutic decisions. This would enhance the practical utility of the review for clinicians.

Response 3: Thank you for your suggestion. We add new table (Table 3) in line 172-173

Round 2

Reviewer 1 Report

Comments and Suggestions for Authors

Table 1 appears to be incomplete in some of the columns (e.g. the word "monosomy" appears to be missing. May bei due to a technical problem with the pdf?

Figure 2, heading: What is meant by "Iter" after diagnostic?? Workflow or algorithm would best  describe the figure

Otherwise, alle comments have been considered.

Author Response

For review article

1. Summary

Thank you very much for taking the time to review this manuscript. In the following section, we have provided responses to the comments that have been raised.

2. Questions for General Evaluation

Reviewer’s Evaluation

Response and Revisions

Is the work a significant contribution to the field?

Is the work well organized and comprehensively described?

Is the work scientifically sound and not misleading?

Are there appropriate and adequate references to related and previous work? 

3. Point-by-point response to Comments and Suggestions for Authors

  1. Table 1 appears to be incomplete in some of the columns (e.g. the word "monosomy" appears to be missing. May bei due to a technical problem with the pdf?

Response 1. The changes and corrections have been made

  1. Figure 2, heading: What is meant by "Iter" after diagnostic?? Workflow or algorithm would best  describe the figure

Response 2: To clarify, we have revised the title of Figure 2 accordingly and modified the relevant section."

Otherwise, alle comments have been considered.

Reviewer 2 Report

Comments and Suggestions for Authors

Dear Authors,

Thank you for addressing the major revisions in your comprehensive review on VEXAS syndrome. The manuscript has been significantly improved through your revisions. However, several minor issues require attention before acceptance.

Minor Revisions Required

  1. Terminology Consistency

The manuscript inconsistently uses "hematopoietic" and "haematopoietic" throughout. Please standardize to one form, preferably "hematopoietic" for consistency with the journal's style.

  1. Figure and Table Improvements

While Figure 2 has been improved as requested, minor enhancements would strengthen the manuscript:

  • Table 1 caption could be more descriptive of the specific findings summarized
  • Consider adding confidence intervals to prevalence data in Table 3 where available
  1. Clinical Guidance Enhancement

The authors have addressed the request for practical clinical guidance. However, consider adding:

  • Specific laboratory value ranges that should trigger VEXAS consideration
  • Brief mention of cost-effectiveness considerations for genetic testing
  1. Gender Analysis Refinement

The enhanced discussion of gender differences is appreciated. However:

  • Consider adding a brief statement about screening recommendations for women with Turner syndrome
  1. Therapeutic Section Organization

The therapeutic approaches section (lines 418-421) could benefit from:

  • More explicit evidence grading for each therapeutic intervention
  • Brief mention of contraindications for key therapies

Conclusion

This manuscript provides valuable insights into VEXAS syndrome and will be a useful resource for clinicians. The minor revisions requested above will enhance clarity and professional presentation. Once these issues are addressed, the manuscript will be suitable for publication.

We look forward to reviewing your revised submission.

Best regards,

Reviewer

Comments on the Quality of English Language

The English could be improved to more clearly express the research.

Author Response

For review article

1. Summary

Thank you very much for taking the time to review this manuscript. In the following section, we have provided responses to the comments that have been raised.

2. Questions for General Evaluation

Reviewer’s Evaluation

Response and Revisions

Is the work a significant contribution to the field?

Is the work well organized and comprehensively described?

Is the work scientifically sound and not misleading?

Are there appropriate and adequate references to related and previous work? 

3. Point-by-point response to Comments and Suggestions for Authors

Thank you for addressing the major revisions in your comprehensive review on VEXAS syndrome. The manuscript has been significantly improved through your revisions. However, several minor issues require attention before acceptance.

Minor Revisions Required

  1. Terminology Consistency

The manuscript inconsistently uses "hematopoietic" and "haematopoietic" throughout. Please standardize to one form, preferably "hematopoietic" for consistency with the journal's style.

Response 1: The term has been standardized to "hematopoietic" throughout the manuscript for consistency with the journal’s style.

  1. Figure and Table Improvements

While Figure 2 has been improved as requested, minor enhancements would strengthen the manuscript:

  • Table 1 caption could be more descriptive of the specific findings summarized
  • Consider adding confidence intervals to prevalence data in Table 3 where available

Response 2: Done

  1. Clinical Guidance Enhancement

The authors have addressed the request for practical clinical guidance. However, consider adding:

  • Specific laboratory value ranges that should trigger VEXAS consideration
  • Brief mention of cost-effectiveness considerations for genetic testing

Response 3: We appreciate the reviewer’s valuable suggestions regarding the inclusion of specific laboratory value ranges and cost-effectiveness considerations. However, we have decided not to incorporate these elements in the current manuscript to maintain its focus and scope. Thank you for your understanding.

  1. Gender Analysis Refinement

The enhanced discussion of gender differences is appreciated. However:

  • Consider adding a brief statement about screening recommendations for women with Turner syndrome

Response 4: This suggestion has been addressed in the revised manuscript, specifically on line160- 166.

  1. Therapeutic Section Organization

The therapeutic approaches section (lines 418-421) could benefit from:

  • More explicit evidence grading for each therapeutic intervention
  • Brief mention of contraindications for key therapies

Response 5: We appreciate the reviewer’s valuable suggestions regarding the inclusion of specific laboratory value ranges and cost-effectiveness considerations. However, we have decided not to incorporate these elements in the current manuscript to maintain its focus and scope. Thank you for your understanding.

Conclusion

This manuscript provides valuable insights into VEXAS syndrome and will be a useful resource for clinicians. The minor revisions requested above will enhance clarity and professional presentation. Once these issues are addressed, the manuscript will be suitable for publication.

We look forward to reviewing your revised submission.

Round 3

Reviewer 2 Report

Comments and Suggestions for Authors

Thank you for addressing the minor revisions requested in our review. After reviewing your responses and the current state of the manuscript, I am pleased to inform you that your revisions are acceptable.

Response Assessment:

Your standardization of terminology to "hematopoietic" throughout the manuscript (Response 1) successfully addresses the consistency concern and aligns with standard journal conventions.

The improvements made to Table 1 caption and related formatting (Response 2) enhance the clarity and accessibility of the presented data.

Regarding the addition of screening recommendations for women with Turner syndrome (Response 4), the inclusion of this guidance in lines 160-166 appropriately addresses the clinical relevance for this patient population.

We acknowledge and respect your editorial decisions regarding Responses 3 and 5, where you chose to maintain the manuscript's current focus and scope rather than incorporating additional elements such as specific laboratory value ranges, cost-effectiveness considerations, evidence grading, and therapeutic contraindications. These decisions are within the authors' purview and do not compromise the manuscript's overall quality or contribution to the literature.

Final Decision:

The manuscript "VEXAS Syndrome: Genetics, Gender Differences, Clinical Insights, Diagnostic Pitfalls, and Emerging Therapies" is now accepted for publication. Your comprehensive review provides valuable clinical insights that will serve as an important resource for healthcare professionals managing this complex syndrome.

We appreciate your thorough attention to the revision process and look forward to seeing this work published.

Congratulations on your successful submission.